# Influence of Maternal Age and Gestational Age on Breast Milk Antioxidants During the First Month of Lactation

**DOI:** 10.3390/nu12092569

**Published:** 2020-08-25

**Authors:** Andrea Gila-Díaz, Gloria Herranz Carrillo, Silvia Cañas, Miguel Saenz de Pipaón, José Antonio Martínez-Orgado, Pilar Rodríguez-Rodríguez, Ángel Luis López de Pablo, María A. Martin-Cabrejas, David Ramiro-Cortijo, Silvia M. Arribas

**Affiliations:** 1Department of Physiology, Faculty of Medicine, Universidad Autónoma de Madrid, C/Arzobispo Morcillo 2, 28029 Madrid, Spain; andrea.gila@uam.es (A.G.-D.); pilar.rodriguezr@uam.es (P.R.-R.); angel.lopezdepablo@uam.es (Á.L.L.d.P.); 2Division of Neonatology, Hospital Clínico San Carlos, Instituto de Investigación Sanitaria del Hospital Clínico San Carlos (IdISSC), C/Profesor Martín Lagos s/n, 28040 Madrid, Spain; gherranz@gmail.com (G.H.C.); jose.martinezo@salud.madrid.org (J.A.M.-O.); 3Department of Agricultural and Food Chemistry-CIAL, Faculty of Sciences, Universidad Autónoma de Madrid, Ciudad Universitaria de Cantoblanco, 28049 Madrid, Spain; silvia.cannas@uam.es (S.C.); maria.martin@uam.es (M.A.M.-C.); 4Department of Neonatology, Hospital La Paz, Paseo de la Castellana 216, 28046 Madrid, Spain; miguel.saenz@salud.madrid.org; 5Division of Gastroenterology, Beth Israel Deaconess Medical Center, Harvard Medical School, 330 Brookline avenue, 02215 Boston, MA, USA

**Keywords:** antioxidant, breast milk, gestational age, maternal age, melatonin

## Abstract

Breast milk (BM) is beneficial due to its content in a wide range of different antioxidants, particularly relevant for preterm infants, who are at higher risk of oxidative stress. We hypothesize that BM antioxidants are adapted to gestational age and are negatively influenced by maternal age. Fifty breastfeeding women from two hospitals (Madrid, Spain) provided BM samples at days 7, 14 and 28 of lactation to assess total antioxidant capacity (ABTS), thiol groups, reduced glutathione (GSH), superoxide dismutase (SOD) and catalase activities, lipid peroxidation (malondialdehyde, MDA + 4-Hydroxy-Trans-2-Nonenal, HNE), protein oxidation (carbonyl groups) (spectrophotometry) and melatonin (ELISA). Mixed random-effects linear regression models were used to study the influence of maternal and gestational ages on BM antioxidants, adjusted by days of lactation. Regression models evidenced a negative association between maternal age and BM melatonin levels (β = −7.4 ± 2.5; *p*-value = 0.005); and a negative association between gestational age and BM total antioxidant capacity (β = −0.008 ± 0.003; *p*-value = 0.006), SOD activity (β = −0.002 ± 0.001; *p*-value = 0.043) and protein oxidation (β = −0.22 ± 0.07; *p*-value = 0.001). In conclusion, BM antioxidants are adapted to gestational age providing higher levels to infants with lower degree of maturation; maternal ageing has a negative influence on melatonin, a key antioxidant hormone.

## 1. Introduction

It is well recognized that breastfeeding has many health benefits for the newborn, promoting development and providing protection against perinatal and long-term morbidities [1,2]. The capacity of breast milk (BM) to prevent diseases is related to the wide range of bioactive factors that it contains—hormones, antioxidants, growth factors and immunoglobulins, among others—which have regulatory and immune functions [3,4]. BM content in nutrients and bioactive molecules is dynamic and it depends on several factors, being the best characterized the period of lactation [3,5]. Maternal variables such as body composition, diet, exposure to toxic substances, maternal age or gestational age may also modify the content of bioactive components in BM. Among these variables, gestational age and maternal age may be important factors. Firstly, preterm birth (before 37 weeks of gestation) is a worldwide public health problem [6,7], being the first cause of death in children under 5 years of age, particularly in low-income countries. Prematurity is also an important problem in high-income countries, linked to the increase in childbearing age and subsequent rise in gestational complications and need of assisted reproduction with multiple pregnancies [8,9]. Prematurity is associated with the development of important morbidities in the first weeks of life, including bronchopulmonary dysplasia, retinopathy of prematurity and necrotizing enterocolitis. These complications are associated with oxidative stress, defined as the imbalance between pro-oxidants (mainly reactive oxygen species (ROS)) and antioxidant defenses [10]. Preterm infants are at higher risk of oxidative stress due to immature antioxidant systems (maturation takes place at the end of gestation), and the frequent use of supplemental oxygen, which increases ROS [11,12,13]. Therefore, it is critical to supply these infants with adequate levels of antioxidants before they develop their own system [12]. In this context, breastfeeding should be considered as the main measure to improve their antioxidant status. The benefits of BM against oxidative stress are demonstrated by the lower content of urine biomarkers in breastfed infants compared to those fed with formula, which has been attributed to the higher and more varied presence of antioxidants in BM [14]. The influence of gestational age on BM antioxidants is controversial. Some reports found a higher antioxidant capacity in preterm infants [15], while others found the opposite trend [16] or no differences [17,18]. The controversy may be related to the fact that, in previous studies, one or two parameters were assessed. BM contains many different antioxidants with synergic functions [4] and a global study addressing different types of antioxidant molecules in BM could provide a better picture.

In high-income countries, prematurity is linked, at least in part, to the increase in childbearing age. The process of ageing is known to decrease antioxidant defenses [19,20,21]; however, whether maternal age influences BM antioxidant levels has not been explored. We hypothesize that BM antioxidant levels are adapted to gestational age and are negatively influenced by maternal ageing. To test this hypothesis, we have analyzed in BM samples different types of antioxidants (small molecules, enzymes, the antioxidant hormone melatonin and total antioxidant capacity) and biomarkers of oxidative damage, at different time points along the first month of lactation.

## 2. Materials and Methods

### 2.1. Population of Study

The population was recruited at the Neonatal Intensive Care Unit (NICU) of Hospital Universitario La Paz (HULP, Madrid, Spain), and at the NICU and Obstetrics and Gynecology service of Hospital Clínico San Carlos (HCSC, Madrid, Spain). This study was approved by HULP and HCSC Ethical Committees (Refs. PI-3063 and 19/393-E, respectively) according to research Helsinki declarations involving human subjects.

Participants were enrolled between 15 June 2018 to 13 March 2020. The inclusion criterion was women with single pregnancy who maintained of lactation during the first month postpartum. As exclusion criteria we included the following fetal or maternal alterations which could affect breastmilk composition: (1) fetal malformations, chromosomal or metabolic abnormality errors, which may affect enzymatic activity or expression, including antioxidant enzymes [22,23], and (2) mothers with Diabetes Mellitus or hypertension-related pregnancy disorders, who may have alterations in plasma antioxidants [24].

Fifty mothers anonymous and voluntary accepted to participate in the study and signed the informed consent. Each participant provided a BM sample at three time points (see below for details), and the following data were recorded from electronic medical data and questionnaires: maternal age (years), origin, educational level, parity, gravidity and gestational age (weeks of gestation). No statistical differences between centers were detected for these variables.

### 2.2. Collection and Processing of Breast Milk Samples

A 5 mL volume of BM was collected at days 7 ± 2, 14 ± 2 and 28 ± 2 of lactation period. It was not possible to get colostrum samples for ethical reason, due to the small volume that can be obtained. BM was collected by each mother by hand self-expression with an electric breast pump (Symphony^®^ Medela, Barcelona, Spain). BM was always collected before feeding the infant and, if available, from both breasts. To collect the samples, the mothers washed their hands and cleaned their breast with a gauze with soap and water. BM collection was performed between 10–12 p.m., and immediately after it was transferred to a glass bottle, previously provided by the hospital staff and stored in the freezer to minimize loss of antioxidants. A.G-D. and D.R-C. were in charge of the whole process, including reminding the mothers about collection, gathering the samples after extraction, and transporting them on ice to the laboratory, where they were immediately processed. The time between extraction and processing took a maximum of 3 h. An aliquot was stored at −80 °C to analyze melatonin, and the rest of the sample was centrifuged three times to obtain the aqueous phase, minimizing turbidity, which could interfere with colorimetric assays. Preliminary experiments demonstrated that the optimal centrifugation protocol was: 1st centrifugation, 2000 rpm, 5 min, 4 °C; 2nd centrifugation 2000 rpm, 10 min, 4 °C; 3rd centrifugation, 2000 rpm, 5 min, 4 °C. Glass serological pipettes were used to extract the aqueous phase in each centrifugation. Thereafter, the processed samples were aliquoted for each assay and stored at −80 °C until used. BM samples were analyzed within a month to minimize loss of antioxidants [25].

### 2.3. Breast Milk Antioxidants and Oxidative Damage Biomarkers

All the colorimetric assays were tested by duplicate. In addition to a blank of the calibration curve, all the assays included an additional blank without the chromophore, to void possible interference of remaining turbidity.

#### 2.3.1. Total Antioxidant Capacity

Total antioxidant capacity of the samples was assessed by the 2,2′-azino-bis (3-ethylbenzothiazoline-6-sulfonic acid) radical cations (ABTS^•+^) method, as previously reported [26]. ABTS^•+^ was obtained by reacting 7 mmol/L ABTS solution with 2.45 mmol/L potassium persulfate and stirring it in the dark at room temperature for 16 h before use. The ABTS^•+^ solution was diluted in 5 mmol/L PBS (pH = 7.4; 1:75; *v*/*v*), to an absorbance of 0.7 ± 0.02 at 734 nm. 30 µL of BM was mixed with 270 µL of ABTS^•+^ solution. The reaction was incubated at 37 °C for 5 min and the absorbance was measured at 734 nm in a microplate reader (Cytation 5; BioTek; Winooski, VT, USA). Calibration curves were constructed using standard solution of Trolox. The ABTS antioxidant capacity was calculated as mg Eq. Trolox/mL.

#### 2.3.2. Thiol Groups

Thiol levels in the sample was quantified using Ellman’s reagent 5, 5-dithiobis-2-nitrobenzoic acid (DTNB) [27], adapted to a microplate reader [28]. The absorbance was measured at 412 nm and thiol levels was expressed as mM Reduced glutathione (GSH)/mL.

*Reduced glutathione (GSH).* GSH levels in breast milk were assessed by a fluorimetric method based on the o-phthalaldehyde reaction [27] adapted to a microplate reader [28]. Fluorescence was measured at 360 ± 40 nm excitation and 460 ± 40 nm emission in a microplate reader (Synergy HT Multimode; BioTek; Winooski, VT, USA). GSH was expressed as mg GSH/mL.

#### 2.3.3. Catalase Activity

Catalase activity was assessed as previously described [28]. This method is based on the oxidation of the Amplex Red (Amplex ultra red reagent; Invitrogen, ThermoFisher, MA, USA) produced by hydrogen peroxide in the presence of horseradish peroxidase and the reduction of fluorescence by catalase present in the sample. BM samples were diluted in H_2_O-Q (1:50; *v*/*v*) and assessment of fluorescence was performed at 530 ± 25 nm excitation and 590 ± 35 nm emission wavelengths in a microplate reader (Synergy HT Multimode; BioTek; Winooski, VT, USA). Catalase activity was expressed as U catalase/mL.

#### 2.3.4. Superoxide Dismutase (SOD) 

SOD activity was assessed by a commercial kit (SOD Activity Assay kit KB-03-011, Bioquochem, Gijon, Spain) according to the manufacturer’s instructions. Briefly, 20 µL of BM was diluted in H_2_O-Q (1:1; *v*/*v*) and added 200 µL of the working solution and 20 µL of Enzymes solution. The reaction was incubated at 37 °C for 20 min. The absorbance was measured at 450 nm in a microplate reader (Synergy HT Multimode; BioTek; Winooski, VT, USA). The SOD activity was expressed in % inhibition.

#### 2.3.5. Melatonin

BM samples were dissolved in distilled H_2_O and filtered under vacuum by a filter (11 µm, Whatman^®^, Sigma-Aldrich, Darmstadt; Germany) for further extraction of melatonin using the solid phase (SPE, cartridge C18, Waters). The eluate was evaporated to dryness under nitrogen gas. The residues were dissolved in distilled H_2_O and melatonin levels were determined by a competitive enzyme immunoassay kit (Melatonin ELISA, IBL-International, Hamburg, Germany) according to the manufacturer’s instructions. The kit is characterized by an analytical sensitivity of 1.6 pg/mL and high analytical specificity (low cross-reactivity). Melatonin levels were expressed as pg/mL.

#### 2.3.6. Lipid Peroxidation

Lipid peroxidation was assessed through evaluation of the levels of malondialdehyde (MDA) and 4-Hydroxy-Trans-2-Nonenal (HNE), which are stable products. MDA + HNE level was determined by a commercial kit (Lipid Peroxidation Assay kit KB-03-002, Bioquochem, Gijon, Spain) according to the manufacturer’s instructions. Briefly, 100 µL of BM was mixed with 325 µL of chromophore reagent kit and incubated at 40 °C for 20 min. 200 µL of the reaction or standard curve samples were transferred to microplate. The absorbance was measured at 586 nm in a microplate reader (Synergy HT Multimode; BioTek; Winooski, VT, USA). MDA + HNE content was expressed as µM.

#### 2.3.7. Protein Oxidation

The levels of carbonylated proteins in BM were measured with the dinitrophenylhydrazine (DNPH)-based method [27], adapted to a microplate reader (Synergy HT Multimode; BioTek; Winooski, VT; USA), measuring absorbance at 370 nm as previously described [28]. The levels of carbonyl groups were determined using extinction coefficient of 2, 4-dinitrophenylhydrazine (ε = 22,000 M/cm) and were expressed as nmol/mL.

### 2.4. Statistical Analysis

Statistical analysis was performed with SPSS software (version 24.0; IBM Company, Armonk, NY, USA) and R software (version 3.6.0, 2018, R Core Team, Vienna; Austria) within R Studio interface using ggpubr, devtools, nlme, car and ggplot2 packages. Data were expressed as median and interquartile range [Q1; Q3]. Kruskal–Wallis for pairwise-comparison test was used to assess differences in BM antioxidants and oxidative damage biomarkers along days of lactation. Rho–Spearman correlation (*ρ*) was used to test the association between BM variables, maternal age and gestational age. Linear regression models with mixed random effects were used to study the influence of maternal age and gestational age on BM antioxidants and oxidative damage biomarkers, adjusted by days of lactation. The random effect (RE) coefficient was calculated to determinate the variability with or within days of lactation. Significance probability was established at *p*-value < 0.05.

## 3. Results

With respect to women origin, the population was composed of 76.0% European, being 72.0% Spanish, 16.0% South American, 2.0% North American, 4.0% North African, and 2.0% Asian. Regarding educational level, 46.0% had university studies, 30.0% had high school education and 22.0% had middle school education. The median of gravity was 2 [1; 3] gestations, and the median of parity was 2 [1; 2] labors. The median of maternal age was 34.0 [31.0; 37.0] years old; the median of gestational age was 33.1 [28.9; 37.9] weeks of gestation. No association was detected between maternal age and gestational age (*ρ* = 0.11; *p*-value = 0.32).

### 3.1. Differences in Breast Milk Antioxidants and Oxidative Damage Biomarkers at Days 7, 14 and 28 of Lactation

The evolution of BM antioxidants and oxidative damage biomarkers at the different time points of lactation are described in Table 1. Regarding antioxidant levels, total antioxidant capacity (measured by ABTS assay) and GSH significantly decreased from day 7 to day 28, while thiol groups, catalase and SOD activities were not significantly different at the 3 points studied. The antioxidant hormone melatonin exhibited a biphasic curve with a peak at the 14th day of lactation.

Regarding oxidative damage biomarkers, lipid peroxidation levels (assessed by MDA + HNE) significantly decreased from day 7 to day 28 of lactation, while protein oxidation (assessed by carbonyl groups) were not modified.

### 3.2. Association between Maternal Age, Breast Milk Antioxidants and Oxidative Damage Biomarkers

Maternal age was not statistically correlated with total antioxidant capacity (Figure 1A). Since ABTS was different along lactation, this correlation was separately analyzed at the three time points, being significantly correlated at day 28 of lactation (*ρ* = −0.54; *p*-value = 0.025). Maternal age tended to be positively correlated with thiol groups and GSH (Figure 1A), being this correlation significant at day 14 of lactation (*ρ* = 0.36; *p*-value = 0.044). Maternal age did not show significant correlations with BM antioxidant enzyme activities (Catalase and SOD). However, a negative significant correlation was found between maternal age and BM melatonin (Figure 1B). We did not detect significant correlations between maternal age and BM oxidation biomarkers (to lipids or proteins), neither globally nor taking the three points of lactation separately (Figure 1C).

### 3.3. Association between Gestational Age, Breast Milk Antioxidants and Oxidative Damage Biomarkers

Gestational age exhibited a negative significant correlation with total antioxidant capacity. No significant correlation was found with thiol groups, while GSH levels showed a negative trend (Figure 2A), being significant at day 14 of lactation (*ρ* = −0.49; *p*-value = 0.005). Gestational age showed a negative significant correlation with SOD activity, while no differences were detected with catalase activity or melatonin levels (Figure 2B). We did not detect a significant correlation between gestational age and BM lipid peroxidation biomarker, neither globally nor taking the three points of lactation separately. However, a negative significant correlation was found between gestational age and BM protein oxidation (Figure 2C).

### 3.4. Linear Regression Models with Mixed Random Effects

Since the day of lactation influenced certain BM antioxidants and oxidative damage biomarkers, to avoid a potential analysis bias, “lactation day” was included as a factor in the regression models. Linear regression models showed that maternal age did not have an influence on BM total antioxidant capacity and GSH levels. However, the model showed that maternal age remained associated with BM melatonin, i.e., the older mother, the lower BM levels (Table 2).

On the other hand, the linear regression models demonstrated the association between gestational age and BM total antioxidant capacity, SOD activity and protein oxidation, i.e., the longer gestational age, the lower antioxidant capacity, SOD activity and protein oxidation in the BM from our cohort (Table 2).

RE coefficients were used to determine the variability of the different parameters analyzed between days of lactation or between mothers within the same day of lactation. Considering maternal age as a predictor, total antioxidant capacity and melatonin levels had a higher variability between days of lactation than between mothers (RE = 0.99 and 0.86, respectively). On the other hand, a higher variability was found between mothers than between days of lactation for GSH (RE = 0.35). Regarding gestational age, total antioxidant capacity, GSH levels, SOD activity and protein oxidation had a higher variability between days of lactation than between mothers (Table 2).

## 4. Discussion

The hypothesis of this work was that BM antioxidant levels are adapted to gestational age and are negatively influenced by maternal ageing. Our main findings, analyzing milk from women at the first month of lactation, evidence that BM antioxidant levels were higher in mothers with shorter gestational age and in the first days of lactation. Regarding the influence of maternal age, although there was some reduction in BM antioxidants, only melatonin level was negatively affected. The adaptation of BM antioxidants to gestational age suggests that mother’s own milk should be provided to premature infants, whenever possible, to reduce their risk of oxidative stress related morbidities. The influence of ageing and period of lactation on melatonin levels, an important antioxidant hormone, should also be taken into consideration; research on the influence of sleep quality on BM melatonin, deserves further attention.

BM antioxidants are important, particularly in the context of prematurity, due to the susceptibility of these infants to oxidative stress. Urinary excretion of hydroxy-deoxyguanosine, a marker of DNA oxidative damage, is lower in BM fed infants compared to those fed with formula [29,30]. Similar results have been found measuring urinary F2-isoprostane [30] or MDA [31], stable biomarkers of oxidative damage to lipids. Moreover, feeding with BM is associated with a lower incidence of a variety of oxidative-stress related illnesses in premature infants [14]. These data provide evidence that BM is protective against oxidative damage in the high-risk population of preterm infants. This effect is likely due to the variety of antioxidants present in BM, which may act in a synergic way to counteract ROS through different mechanisms, thus providing a better protection than formula milk, which only contains vitamins, but lacks other antioxidants, such as enzymes, present in BM [32,33]. In this context, it is important to analyze if BM antioxidants are adapted to the needs of the infants, assessing if they are modified with gestational age and along lactation period.

It is well established that macronutrient content is modified along lactation. Proteins, are highest in colostrum, while fat and carbohydrates increases along lactation [34,35]. Similarly, colostrum has been reported to have higher content of antioxidants compared to mature milk [16,36]. Although we could not assess antioxidant levels in colostrum, due to ethical restrictions, our data in transition milk (7 and 14 days) and mature milk (28 days) are in agreement with a gradual decrease of antioxidant capacity along the first month of lactation. Similar results using FRAP technique, other method to assess antioxidant capacity, have been described by Oveisi et al. in Iranian women [37]. RE coefficients were used to determine variability between days of lactation or between mothers within the same day of lactation. The high RE for ABTS confirmed that the variability could be attributed to the period of lactation rather than to differences between mothers. The higher antioxidant capacity at the beginning of lactation is likely a physiological response providing the maximum antioxidant levels during the critical period of transition from intrauterine life to a high oxygen environment. It is important to note that total antioxidant capacity is a parameter which reflects the global capacity of BM and its various antioxidants. From the molecules assessed in the present study, only GSH followed a similar trend, while no modification along the lactation period was observed for the enzymatic antioxidants. Therefore, our data suggest that GSH is an important contributor to total antioxidant capacity of BM.

Melatonin is a key antioxidant enzyme, with multiple actions, as direct and indirect antioxidant. We found that melatonin followed a biphasic trend with a peak at day 14. The concentrations found in our study were in the range of those found in other studies and even higher. The study of Qin et al. demonstrated that melatonin declined the along lactation period with the highest concentration in colostrum [38]. Katzer et al. evidenced higher concentrations at night in both term and preterm milk [17]. This indicates a similar circadian pattern in plasma and BM. Plasma melatonin levels are known to be disrupted with frequent awakenings, and a disturbed sleep patterns occur during the first 7 days postpartum, with improvement around day 20 [39]. Therefore, it is possible to suggest lower melatonin BM in women with poorer sleep. We are aware that this assumption is speculative and assessing the relationship between sleep hygiene and melatonin levels in BM deserves a specific well-designed study.

Regarding the relationship between oxidative stress biomarkers and the lactation period, we found a decrease in lipid peroxidation along the first month. This is in contrast with the findings of Yuksel et al. who found that the lipid peroxidation levels where higher in mature milk compared to colostrum and transitional milk [40]. The content of lipids increases along lactation [35], particularly polyunsaturated fatty acids [41], which are more likely to be oxidized. Accordingly, lipid peroxidation could also increase. Our findings of a reduction of biomarkers of oxidative damage to lipids along the first month of lactation suggest a concomitant reduction in ROS in BM. ROS, such as H_2_O_2_, are synthetized by the lactocytes in the mammary gland [32]. In low concentrations they have actions as antimicrobial agents, but at higher levels they may be harmful as oxidant molecules [42]. It has been demonstrated that BM hydrogen peroxide peaks within days after birth, and decline towards the fourth postnatal week [43]. Therefore, it is possible that the lower MDA + HNE levels observed along lactating could be related to a decrease in ROS production in the mammary gland. Besides, it is possible that other antioxidants not assessed in this study, such as vitamin E and D, protects against lipid peroxidation. Vitamins are highly influenced by maternal diet, and it is possible that modification in maternal diet along lactation could influence the results. As we propose with sleep hygiene, it is possible that maternal diet is modified along lactation, particularly in women with preterm infants, who spend time at the NICU and may not be able to consume their regular diet. Maternal nutrition is an important aspect influencing BM, which was not explored in this study, and deserves further attention.

Some studies have shown that macronutrient of BM changes in response to many factors, mainly lactation period, but also gestational age [44,45]. We propose that gestational age may be an important determinant of BM antioxidant levels, which may be adapted, being higher in preterm infants matching their higher needs. Data regarding the influence of prematurity on BM antioxidants is controversial, with some reports showing that BM from preterm infants has higher levels [15], while others show lower [16] or no differences [17,18], compared to term infants. This controversy may be related to different antioxidants analyzed in each study. Besides, it can be related to differences in gestational ages analyzed, which in preterm infants may range from 24 ^+ 0^–36 ^+ 7^ weeks, or even lower (as low as 22 weeks), although in our study, we did not have infants below 24 weeks. Therefore, we approached the problem analyzing the relationship between different antioxidants and gestational age, rather than comparing term with preterm. We found a negative correlation between most antioxidants in BM and gestational age, which would support the hypothesis that BM is adapted to gestational age, providing more antioxidants to children with lower maturation. Although a trend was found for most antioxidants analyzed, the linear regression model evidenced that only total antioxidant capacity and SOD activity were negatively associated with gestational age. The negative association found in our study is in contrast with the report of L’Abbe and Friel, who found that SOD activity was higher in full term compared to preterm milk [46]. This result may be related to the quantification of SOD in relation to protein level, which was larger in BM from preterm infants. In fact, when quantified as U SOD/mL, no differences were found. In the same study, glutathione peroxidase quantified as mU/mL, was higher in preterm milk in the first weeks of lactation and similar findings were reported in other study at day 21 [47]. We did not find an association between gestational age and melatonin levels at the time points analyzed. This is in agreement with a previous study which reported no differences in melatonin between preterm and term transitional or mature milk [38]. However, the same study found higher melatonin in preterm BM colostrum, suggesting a relationship between antioxidant levels in BM matching infant needs [38].

Regarding oxidative stress biomarkers in milk, we found that carbonyl groups (protein oxidation biomarker) in BM, exhibited a negative correlation with gestational age. This result can be explained by the decrease protein levels in BM along lactation period [44,48]. However, it is also possible that GSH and SOD may be important in the protection of BM proteins from oxidation, since they followed a parallel relationship with gestational age. Instead, lipid peroxidation biomarkers were not influenced by gestational age. It is possible that other antioxidants not assessed in this study, such as vitamin E or D, exert more important antioxidant actions on lipids. These vitamins may not be influenced by gestational age, being mainly dependent on maternal sources, either their diet or lipid deposits acquired during pregnancy.

Regarding the influence of maternal age on antioxidants, we found that only melatonin levels were significantly affected during the first month of breastfeeding, also confirmed by the linear model. Maternal age is one of the main factors limiting antioxidant production in humans, as established by previous studies, and extensively demonstrated for melatonin [49,50]. Melatonin levels in plasma are known to decline with age [51] and plasma concentrations match those found in BM [38]. These data suggest that pregnancies with advance maternal age, defined as age above 35 years old [52], could be more susceptible to low levels of melatonin in BM. In addition to the antioxidant effects, other effects described of melatonin are relaxing on gastrointestinal smooth muscle and a hypnotic effect [53]. It is possible that maternal melatonin passes into the BM to the infant could contribute to improve infant sleep, early establishment of circadian rhythm and to decrease infantile colic. Given the importance of this hormone as antioxidant and the stimulation of other antioxidants, the modifications in BM melatonin with age and lactation period, reported in the present study, which may be influenced by maternal sleep deserve further analysis.

### Study Limitations and Strengths

One of the limitations of the study could be the sample collection methodology, since only one sample was obtained at a single time point during the day. Therefore, information about changes with circadian rhythms, which may influence some of the BM parameters analyzed [54], particularly melatonin and some antioxidant enzymes, has not been possible. The exclusion of some important antioxidants, such as vitamins, carotenoids or coenzyme Q, could be considered a potential pitfall. However, we did not assess them since there is evidence that their content in BM is markedly influenced by maternal diet [55,56,57].

Furthermore, it would be interesting to explore the relationship between antioxidants in breast milk and neonatal variables, such as health status and morbidities, as well as to test how these bioactive factors of the BM respond to neonatal physiology.

On the other hand, the collection in two centers including women of different ethnic and socioeconomic background is a strong point of the study, supporting the validity of our findings regarding maternal biological factors. Another strength is the careful treatment of the sample and quantification protocols. We used a quick processing time to prevent the degradation of antioxidants, which may occur when milk is stored for 24 h in the refrigerator [58,59]. We also improved protocols to avoid turbidity interference, including three-steps of centrifugation and an additional sample blank.

## 5. Conclusions

Breastmilk antioxidants are reduced along lactation period, probably reflecting lower needs with infant development and adaptation to extrauterine environment. The negative relationship between breastmilk antioxidants and gestational age suggests an adaptation, providing higher levels to infants with lower degree of maturation, and supports the view that premature newborns will be better protected with breast milk from their mothers. Maternal age does not compromise overall antioxidant levels in breastmilk, with the exception of melatonin. Given the important role of this neurohormone as antioxidant and sleep inducing substance, this fact could be taken into account for women with pregnancy in advance maternal age.

## Figures and Tables

**Figure 1 nutrients-12-02569-f001:**
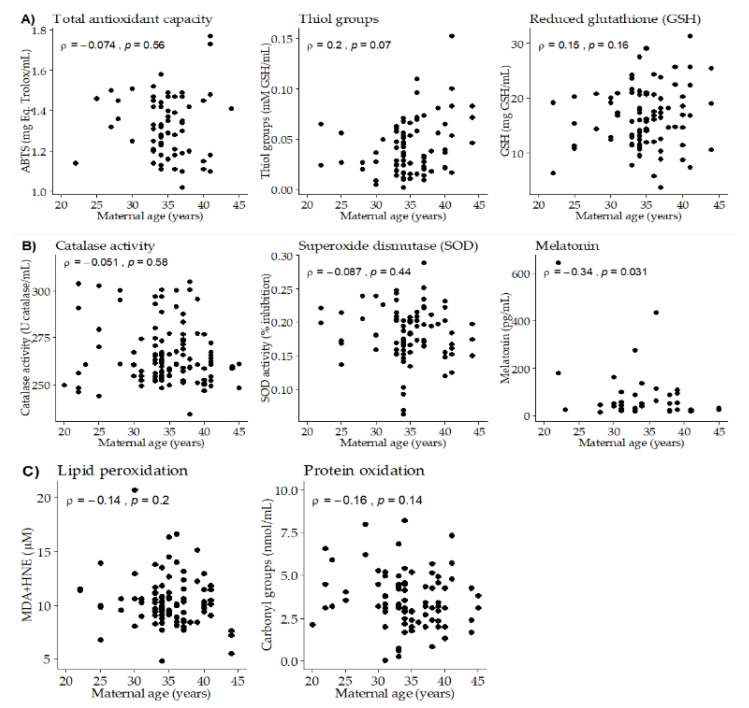
Scatter plots of breast milk antioxidants and oxidative damage biomarkers versus maternal age. (**A**) Total antioxidant capacity and low molecular weight antioxidants. (**B**) Antioxidant enzymes. (**C**) Oxidation biomarkers. Rho–Spearman test (*ρ*) and associated *p*-values are shown in each plot.

**Figure 2 nutrients-12-02569-f002:**
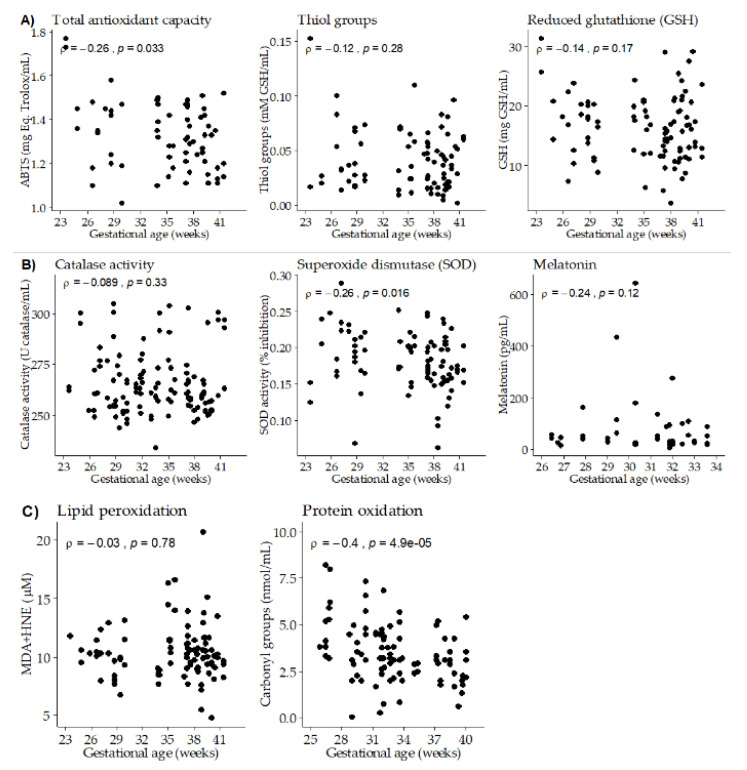
Scatter plots of breast milk antioxidants and oxidative damage biomarkers versus gestational age. (**A**) Total antioxidant capacity and low molecular weight antioxidants. (**B**) Antioxidant enzymes. (**C**) Oxidation biomarkers. Rho–Spearman test (*ρ*) and associated *p*-values are shown in each plot.

**Table 1 nutrients-12-02569-t001:** Breast milk antioxidants and oxidative damage biomarkers at days 7, 14 and 28 of lactation.

	Day 7 (*n* = 50)	Day 14 (*n* = 50)	Day 28 (*n* = 45)	*p*-Value
ABTS (mg Eq. Trolox/mL)	1.45 (1.35; 1.48)	1.30 (1.18; 1.37)	1.20 (1.11; 1.25)	< 0.001
Thiol groups (mM GSH/mL)	0.04 (0.02; 0.06)	0.03 (0.02; 0.06)	0.04 (0.02; 0.06)	0.85
GSH (mg GSH/mL)	20.2 (15.8; 22.5)	16.2 (12.1; 18.4)	12.2 (10.5; 15.8)	< 0.001
Catalase activity (U/mL)	261.5 (256.6; 271.1)	261.3 (253.6; 277.0)	262.0 (253.5; 274.7)	0.94
SOD activity (% inhibition)	0.17 (0.15; 0.20)	0.17 (0.16; 0.22)	0.19 (0.17; 0.20)	0.95
Melatonin (pg/mL)	25.9 (17.2; 49.7)	70.7 (26.3; 129.7)	31.3 (23.2; 51.1)	0.07
MDA + HNE (µM)	10.6 (9.7; 12.1)	10.3 (9.2; 11.7)	9.3 (8.3; 10.0)	< 0.001
Carbonyl groups (nmol/mL)	3.7 (2.3; 4.2)	3.3 (2.0; 4.0)	3.1 (2.0; 3.1)	0.66

Data show median and interquartile range (Q1; Q3). ABTS: 2, 2′-Azino-Bis-3-Ethylbenzothiazoline-6-Sulfonic Acid; GSH: reduced glutathione; SOD: Superoxide Dismutase; MDA: Malondialdehyde; HNE: 4-Hydroxy-Trans-2-Nonenal. *p*-value by Kruskal-Wallis test.

**Table 2 nutrients-12-02569-t002:** Linear regression models with mixed random-effect by day of lactation.

	Maternal Age	*p*-Value	RE	Gestational Age	*p*-Value	RE
ABTS (mg Eq. Trolox/mL)	0.001 ± 0.004	0.78	0.99	−0.008 ± 0.003	0.006	0.99
GSH (mg GSH/mL)	0.19 ± 0.12	0.11	0.35	−0.13 ± 0.10	0.22	0.94
SOD (% inhibition)	-		-	−0.002 ± 0.001	0.043	0.97
Melatonin (pg/mL)	−7.41 ± 2.50	0.005	0.86	-	-	-
Carbonyl groups (nmol/mL)	-		-	−0.22 ± 0.07	0.001	0.93

Data show the estimated β ± standard error and *p*-value associated. Random effect (RE) calculated as variance between days of lactation over total variance of the model. ABTS: 2, 2′-Azino-Bis-3-Ethylbenzothiazoline-6-Sulfonic Acid; GSH: reduced glutathione; SOD: Superoxide Dismutase.

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
