# Peer review of "Influence of Maternal Age and Gestational Age on Breast Milk Antioxidants During the First Month of Lactation"

_nutrients, 2020, doi:10.3390/nu12092569_

Round 1
Reviewer 1 Report
The study is interesting and provides new data on breast milk. The description of the examined group of women and the inclusion and exclusion criteria as well as the methodology of milk collection require significant correction and clarification.
84-87 the inclusion and exclusion criteria must be clearly defined, which means "without any ... chromosomal or metabolic abnormality errors, which may affect enzymatic activity, and maintenance of lactation during the first month..." You have included premature pregnancies but have you analyzed what was the reason of prematurity. Have you excluded diabetes, hypertension or PPROM?
93-95 The method of milk extraction was not described. The variability of milk during the day is so high that it can have a significant influence also on the antioxidant activity of milk. Similarly to expressing milk before or after feeding the baby, or obtaining from the whole pumped portion, some of which is given to the baby and only the residue is used for testing. why was the milk pumped between 10 and 12? How could this time of day be justified? How and who collected milk from women on the 14th and 28th day of lactation? Were the women patients in the hospital at that time?
170-175 The characteristics of the examined female population should be included in the material. No data on newborns are available. Were they breastfed or received their own mother's suckling milk?
Minor corrections and clarifications are required for unprofessional perinatal medical terms
54 "important morbidities in the immediate neonatal period" must be corrected what means immediate neonatal period
220 "the larger the age" should be corrected
222-223 "the larger the gestational age, the lower the antioxidant capacity" also should be changed
239-240 "lower gestational age" it also should be changed
260-261 "Although we could not assess antioxidant levels in colostrum, due to ethical reasons" it is a pity that you did not use lactation consultants, in many publications, thanks to their work, it was possible to obtain colostrum according to the rules of ethics. It is necessary to complete the method of milk collection to study who educated mothers, who supported them in stimulating lactation
278-281 "We suggest that the peak at day 14 could reflect changes in sleeping patterns in women, since this neurohormone follows circadian rhythms also in BM with higher nighttime concentration [17]. It is possible that the higher concentrations observed at day 14 may reflect a better sleep hygiene of the mother." It must be explain. Have you examined mothers milk obtained during night time? Is the conclusion the effect of your study?
301 „Besides, it can be related to differences in gestational ages analyzed, which in preterm infants may range from 24+0-36+7 weeks” why 24, it should be 22
302-303 „…the relationship between different antioxidants and gestational age, rather than comparing term with preterm „ it should be changed in your study you have analyzed the milk of women who gave birth prematurely and on time, as determined by their gestational age. it is always necessary to determine the gestational age to say preterm or term.
326 „advance maternal age” it is worthwhile to determine the age in years, what means advance?
354 „older childbearing age” as above it is worthwhile to determine the age in years, what means older?
English, especially concerning medical terminology in perinatology, needs to be corrected.
Author Response
The study is interesting and provides new data on breast milk. The description of the examined group of women and the inclusion and exclusion criteria as well as the methodology of milk collection require significant correction and clarification.
Response: We would like to thank the reviewer for the time spent on our manuscript as well as for the comments and suggestions made. We hope that we have kindly answered all the queries.
84-87 the inclusion and exclusion criteria must be clearly defined, which means "without any ... chromosomal or metabolic abnormality errors, which may affect enzymatic activity, and maintenance of lactation during the first month..." You have included premature pregnancies but have you analyzed what was the reason of prematurity. Have you excluded diabetes, hypertension or PPROM?
Response: Our exclusion criteria was based on those alterations, which may affect breast milk antioxidant composition and we are aware that it was not explicitly defined in the manuscript. We have now clarified this in the text (lines 85-90). On one hand, we excluded mothers with neonates with chromosomal or metabolic abnormality errors, since they could affect the level or activity of endogenous enzymes, including antioxidant enzymatic systems, which could bias our results. For example, the gene coding for the enzyme SOD is localized to 21q22.1 and patients with trisomy 21 could be at risk to overexpress this antioxidant enzyme (PMID: 6217779; PMID: 25852816). On the other hand, women with gestational diabetes or pregnancy induced hypertension were also excluded, since these alterations affect their antioxidant levels in plasma (PMID: 27780538), and we cannot exclude breastmilk antioxidants can be affected.
We included women with premature infants, since we think it is important to gain knowledge on the level of antioxidants according to gestational age. Unfortunately, we did not record the reasons of prematurity from medical records.
93-95 The method of milk extraction was not described.
The variability of milk during the day is so high that it can have a significant influence also on the antioxidant activity of milk. Similarly, to expressing milk before or after feeding the baby, or obtaining from the whole pumped portion, some of which is given to the baby and only the residue is used for testing. why was the milk pumped between 10 and 12? How could this time of day be justified? How and who collected milk from women on the 14th and 28th day of lactation? Were the women patients in the hospital at that time?
Response: We are aware of the great variability of BM along the day, time of lactation and other important factors which may influence its components, questions raised by the reviewer, which were not included in the manuscript. We have now added this information in the new version:
1) all the process was conducted by 2 researchers who contacted the mother before the collecting time, went to pick up the sample and processed it
2) the milk was obtained before feeding the baby (if possible from both breasts)
3) we, arbitrarily, decided to collect between 10-12h because was a convenient time for mothers and researchers (they had to phone to remind the mother about the collection, go to collect the sample and process it)
4) regarding the place of collection, some women were in the hospital during study (women with preterm infants or morbidities), or at their homes (hospital discharged 72h for full-term deliveries, according to medical-staff guidelines). Al these details regarding breast milk extraction are now included in the text (lines 97-107).
We could not collect milk at different time points along the day for ethical reasons (particularly for mothers with preterm infants); we agree on the importance of this aspect and included it as a limitation.
170-175 The characteristics of the examined female population should be included in the material. No data on newborns are available. Were they breastfed or received their own mother's suckling milk?
Response: The maternal variables (age, origin, educational level, parity, gravidity and gestational age) are included in the material and method section (lines 93-94).
Neonatal variables were not included because the main focus of the manuscript was to explore the antioxidant levels in breast milk and their relationship with maternal factors. The study design did not allow assessing the relationship between milk components and neonatal parameters, such as growth, which we understand are very important. Since we were interested to explore the influence of gestational age, and premature infants were not only fed on their own milk, we decided not to explore this aspect which requires a different experimental design. It would be interesting to explore the impact of BM antioxidants and neonatal parameters, and it will be the next research step. We have included the research gap in the discussion section (lines 403-405).
Regarding to the way of breastfeeding, infants from full-term deliveries were directly breastfed, although in some instances the mother obtained the milk with a pump. Instead, mothers with premature delivery obtained the milk with a pump, since the infants were not able to suckle, and they received their own mother´s milk together with donated milk, according to Neonatal Intensive Care Unit state guidelines.
Minor corrections and clarifications are required for unprofessional perinatal medical terms
54 "important morbidities in the immediate neonatal period" must be corrected what means immediate neonatal period.
Response: we have changed the sentence to improve the understanding and corrected some perinatal terms.
220 "the larger the age" should be corrected
Response: We have corrected as “the older the mother, the lower BM levels”
222-223 "the larger the gestational age, the lower the antioxidant capacity" also should be changed
Response: We have corrected as “the longer the gestational age, the lower the antioxidant capacity”
239-240 "lower gestational age" it also should be changed
Response: We have changed the sentence to “with shorter gestational age”.
260-261 "Although we could not assess antioxidant levels in colostrum, due to ethical reasons" it is a pity that you did not use lactation consultants, in many publications, thanks to their work, it was possible to obtain colostrum according to the rules of ethics. It is necessary to complete the method of milk collection to study who educated mothers, who supported them in stimulating lactation
Response: As the reviewer mentions, it would be very interesting to be able to explore these components in colostrum. Unfortunately, we could not do so, due to the restrictions in mothers of preterm infants, who reserve all their milk for the babies. At the hospitals where the samples were obtained, the neonatologists collaborate with lactation consultants, who are in charge of women advice. However, the consultants provide their services only on demand of the mothers and not on usual. Besides, the small volume that can be obtained of colostrum is reserved always for the neonate, and some mothers did not have sufficient. Later on, when mothers are discharged from hospitals, the infant is followed at Primary Health Care Centers, where the pediatrician and midwife are in charge of breastfeeding education.
278-281 "We suggest that the peak at day 14 could reflect changes in sleeping patterns in women, since this neurohormone follows circadian rhythms also in BM with higher nighttime concentration [17]. It is possible that the higher concentrations observed at day 14 may reflect a better sleep hygiene of the mother." It must be explain. Have you examined mothers milk obtained during night time? Is the conclusion the effect of your study?
Response: We did not collect BM during night, but other authors have demonstrated a circadian pattern of BM melatonin with higher concentrations at night, in both term and preterm milk (PMID: 27121237). This indicates that there is a similar pattern of melatonin concentration in plasma and BM. Plasma melatonin levels are known to be disrupted with frequent awakenings, and a disturbed sleep pattern has been demonstrated during the first 7 days postpartum, with improvement and establishment of a normal pattern by day 20 (PMID: 19208049). Therefore, it is possible to suggest lower melatonin BM levels in women with poorer sleep. However, we believe this assumption is speculative and assessing the relationship between sleep hygiene and melatonin levels in BM deserves a specific well-designed study. We have, added a sentence regarding this aspect in discussion (lines 323-328)
301 “Besides, it can be related to differences in gestational ages analyzed, which in preterm infants may range from 24+0-36+7 weeks” why 24, it should be 22
Response: In our study, the youngest infant was born at 23+6 weeks of gestation, so we did not have extremely premature infants. The Neonatal Research Network estimates a 6% of survival of infants born with 22 weeks of gestation; accordingly, they no recommend any interventional studies with this infant group (PMID: 30478270). We have added a sentence in the discussion about limits of gestational age (line 356).
302-303 “…the relationship between different antioxidants and gestational age, rather than comparing term with preterm” it should be changed in your study you have analyzed the milk of women who gave birth prematurely and on time, as determined by their gestational age. it is always necessary to determine the gestational age to say preterm or term.
Response: We agree on the fact that gestational age defines the delivery as term or preterm and many studies have approached the influence of prematurity on BM components. However, since the word preterm includes neonates of very different gestational ages, we decided to explore gestational age variable as a continuous variable, as opposed to studies in which establish a cut-off point of gestational age as less than 37 weeks to define preterm birth.
326 “advance maternal age” it is worthwhile to determine the age in years, what means advance?
Response: According to a recent meta-analysis in pregnancies outcomes (PMID: 30946794), the most fertile period of women is between 20 and 34 years. Thus, advanced maternal age is considered as maternal age to gestation above 35 years. We have modified the text (line 387) to include the data.
354 “older childbearing age” as above it is worthwhile to determine the age in years, what means older?
Response: we have re-write as “…this fact could be taken into account for women with pregnancy in advance age, i.e. women above 34 years of age” to emphasize that, according to our results, breast milk melatonin decreased with each year of age of the mothers.
English, especially concerning medical terminology in perinatology, needs to be corrected.
Response: English has been revised, with special attention to medical terms.
Reviewer 2 Report
More could be added to the significance of the findings in the discussion, since so much work was done, but little is discussed about the implications
minor typo: line 329, "maternal melatonin massing through milk"
Author Response
Response: We would like to thank the reviewer for the time spent on our manuscript as well as for the comments and suggestions made.
More could be added to the significance of the findings in the discussion, since so much work was done, but little is discussed about the implications.
Response: Thank you for your comments. We have no extended the discussion including some aspects which were not included in our previous versions, including additional discussion on melatonin and sleep, relationship between antioxidants and markers of oxidative damage and the influence of maternal diet.
minor typo: line 329, "maternal melatonin massing through milk"
Response: We have corrected it.